# Effects of Residual Stresses on the Fatigue Lifetimes of Self-Piercing Riveted Joints of AZ31 Mg Alloy and Al5052 Al Alloy Sheets

**Young-In Lee [1] and Ho-Kyung Kim [2],***

1 Graduate School, Seoul National University of Science and Technology, Seoul 01811, Korea; yilee1234@seoultech.ac.kr
2 Department of Mechanical and Automotive Engineering, Seoul National University of Science and Technology, Seoul 01811, Korea
* Correspondence: kimhk@seoultech.ac.kr; Tel.: +82-2-970-6348

**Abstract:** During the self-piercing riveting (SPR) process, residual stress develops due to the high plastic deformation of the sheet materials. In this study, the effect of the residual stress on the fatigue lifetime of SPR joints with dissimilar magnesium AZ31 alloy and aluminum Al5052 alloy sheets was evaluated. The residual stress distribution was derived through a simulation of the SPR process by the FEA (finite element analysis). The measured values by the X-ray diffraction technique confirmed that the validity of the simulation has a maximum error of 17.2% with the experimental results. The fatigue strength of the SPR joint was evaluated at various loading angles using tensile-shear and cross-shaped specimens. It was found that the compressive residual stresses of the joint reduce the stress amplitude by 13% at $10^6$ cycles lifetime, resulting in extension of its lifetime to approximately 3.4 million cycles from $10^6$ cycles lifetime. Finally, it was confirmed that the fatigue life of SPR joints was appropriately predicted within a factor of three using the relationship between the fatigue life and the equivalent stress intensity factor. The fatigue resistance of the magnesium AZ31 alloy on the upper sheet was found to govern fatigue lifetimes of SPR joints of dissimilar magnesium AZ31 alloy sheets.

**Keywords:** self-piercing riveting; magnesium alloy; fatigue strength; equivalent stress intensity factor; fatigue lifetime

## 1. Introduction

In order to decrease carbon emissions and improve fuel efficiency outcomes, the automobile industry is taking the approach of reducing vehicle body weights using high-specific-strength materials, including aluminum, magnesium, and CFRP (carbon-fiber-reinforced plastic) [1]. Increasing the percentage of high specific strength alloys, e.g., aluminum alloys and magnesium alloys makes it possible to achieve vehicle body weight reductions while maintaining structural stiffness and strength. The combination of aluminum alloys and magnesium alloys offers automotive designers the possibility to optimize the body structure in terms of weight reduction and performance [2]. However, the conventional resistance-spot-welding method used to join steel plates cannot be used to join these non-ferrous materials, due to significant differences in the melting temperature and electrical conductivity. The self-piercing riveting (SPR) method has become a promising alternative joining method for non-ferrous material plate joining [3,4]. A schematic of an SPR joint is shown in Figure 1. SPR is a high-speed mechanical joining technique, during which a semi-tubular rivet is pressed by a punch into two plates of materials that are supported on a die. The rivet pierces the top plate, partially piercing the bottom plate. The rivet tail flares within the bottom plate to create a mechanical interlock under the supervision of an appropriate die shape. The strength of an SPR joint is primarily determined by the interlock.

Compared to more traditional joining methods, such as blind riveting, spot-welding, and adhesive bonding, other advantages of SPR include environmental friendliness and the achievement of high joining strength levels [3].

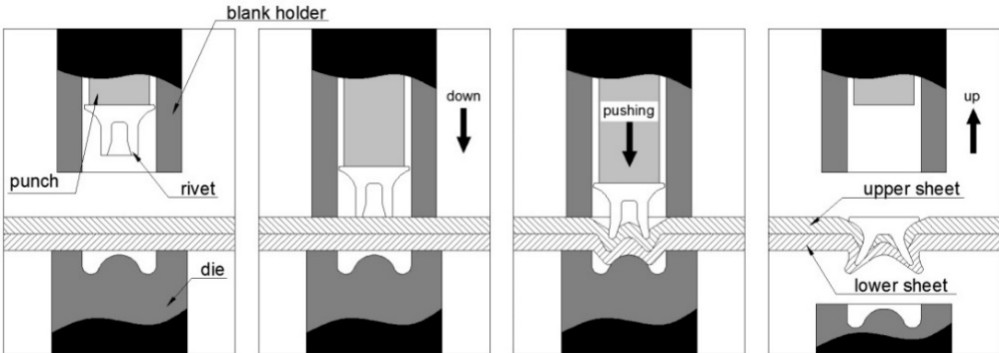

**Figure 1.** Illustration of the self-piercing riveting process.

Due to the increasing adoption of the SPR method, several studies have examined the static and fatigue strengths of SPR joints with various material plates [5–10]. Kawamura and Cheng proposed a method to predict the S-N curves of SPR joints with aluminum alloy plates using the stress intensity factor based on the RPG (re-tensile plastic zone generating) load [5]. In order to predict the fatigue lifetime of SPR joints in tensile-shear and coach-peel geometries, they measured the fatigue crack growth rates, calculated the stress intensity factors and estimated the crack paths of the fatigue specimens through the FEA, eventually obtaining the S-N curves of the specimens. They argued that their proposed methodology is effective and efficient for predicting S-N curves. Rao et al. studied the fatigue properties of SPR joints with CFRP and aluminum alloy plates in lap-shear and cross-tension specimens [6]. They adopted a structural stress model to predict the fatigue lifetimes of SPR joints for both specimen types. They reported that in fatigue tests, the lap-shear joints failed due to kinked crack growth in the bottom aluminum plate, while the cross-tension joints failed when the rivet pulled out of the top CFRP plate. They stressed the importance of the flared rivet diameter as a controlling parameter when evaluating the fatigue lifetimes of the SPR joints. Su et al. studied the fatigue strength of SPR joints with aluminum alloy plates in tensile-shear specimens [7]. They proposed to estimate the fatigue lifetime of a joint by adopting structural stress solutions at the crack initiation sites and the stress-life data of the specimen material. They showed that their proposed fatigue lifetime estimations were in good agreement with experimental data. Zhao et al. investigated the fretting behavior of SPR joints in titanium plates during fatigue tests [8]. They reported that pierced plate failures were mainly caused by fretting wear at the interface between the rivet head and the pierced plate. In addition, locked plate failures and rivet failures were generated by fretting wear at the interface between the rivet shank and the locked plate. Kang et al. investigated the static and fatigue strength levels of SPR joints of magnesium alloy and cold-rolled steel plates under various loading conditions using a cross-shaped type of specimen geometry [9]. They reported that the fatigue ratios at loading angles of 0°, 45°, and 90° are 22%, 13%, and 9%, respectively. They argued that the fatigue lifetimes could be evaluated most appropriately through the maximum principal stress. Most of these studies have evaluated the fatigue strengths of the SPR joint specimens using load amplitude-lifetime curves instead of conventional stress–lifetime curves. However, it is difficult to predict the fatigue lifetimes of the specimens if the geometry and size of the specimens are different. Few studies have evaluated the fatigue strengths of SPR joints using structural parameters.

During the SPR process, residual stress develops due to the high plastic deformation of the rivet and plate materials [11,12]. In order to obtain information about the residual stress distribution of SPR joints, many researchers adopt finite element analysis (FEA) or/and

X-ray diffraction measurements in various materials [13–19]. For example, Haque et al. proposed to measure residual stresses in SPR joints by a neutron diffraction technique [13]. They observed tensile stress in the rivet head for a steel–steel joint. They reported that the maximum value of the compressive residual stress was 550 MPa and that it occurred in the rivet tail of an aluminum–steel joint. Haque et al. also measured the residual stress profiles in two different SPR joints using the neutron diffraction technique [14,15]. They reported that the residual stress in the plate material inside the bore of the rivet was compressive at the center and that the stress gradually became tensile away from the center. They also reported that the compressive residual stress in the rivet tail was greater for a thin joint than for a thick joint due to the higher force gradient encountered during the rivet flaring stage. Huang et al. conducted a two-dimensional axisymmetric finite element analysis to simulate the SPR process with aluminum alloy and high-strength steel plates [16]. They adopted an element erosion technique in an explicit analysis of the piercing of the top plate with several fracture criteria. The maximum shear strain criterion was proven to be the best way to describe the material separation during the riveting process. The stress distribution was found to change greatly before and after spring-back. They reported that a large amount of compressive residual stress was distributed in the top aluminum plate, which could extend the fatigue lifetime of the SPR joint. Huang et al. also conducted a simulation of the SPR process and measured the residual stress distribution of SPR joints using neutron diffraction measurements [17]. They reported that good agreement was found between the simulation and residual stress measurements. Moraes et al. conducted a three-dimensional and explicit finite element analysis to simulate the SPR process with magnesium and aluminum alloy plates [18]. A stress triaxiality-based damage material model was adopted to model the piercing of the plate by the rivet. They also conducted X-ray diffraction measurement to obtain the distribution of residual stresses in SPR joints. FEA was shown to be capable of predicting the deformed geometry of an experimental cross-section. They reported that the residual stress distribution by FEA correlated well with the trends and magnitudes of the experimentally measured residual stresses.

The effect of residual stress on the fatigue lifetime of a component can be either beneficial or harmful, depending on its magnitude and distribution. Generally, the high-cycle fatigue lifetime of a welded joint decreases due to tensile residual stress near a fatigue crack of the weldment by increasing the mean stress of cyclic loading [19,20]. Thus far, very limited work on the effect of residual stress on the fatigue strength of SPR joints is available in the published research [21]. For example, Jin and Mallick [21] reported that the fatigue lifetimes of SPR joints with aluminum Al5754 alloy plates can be enhanced significantly by introducing compressive residual stress around the coined area through a coining process. However, they did not report any experimentally measured values of the residual stress in their study.

Therefore, in this study, the effect of residual stress on the fatigue lifetimes of SPR joints with dissimilar magnesium and aluminum alloy plates was evaluated. The residual stress distribution was derived through a simulation of the SPR process by the FEA (finite element analysis). The residual stresses of the joints were also measured using the X-ray diffraction technique in order to confirm the validity of the SPR process simulation. The fatigue strengths of SPR joints were evaluated at various load conditions using various fatigue parameters to derive an appropriate structural parameter for predicting their fatigue lifetimes. Finally, the effects of residual stress on the fatigue strength were quantitatively evaluated by comparing these fatigue strengths of the joints with and without consideration of the residual stresses.

## 2. Experimental and FEA Procedures

### 2.1. Materials and Specimens

In this study, the materials used for the SPR joint specimens were magnesium alloy (AZ31) plates and aluminum alloy (Al5052) plates with thicknesses of 1.5 mm. The mechanical properties of these two materials are summarized in Table 1. To evaluate the static

and fatigue strengths of the SPR joints, tensile-shear and cross-shaped specimens, as shown in Figure 2, were prepared. For the cross-shaped SPR specimens at loading angles of 45° and 90°, a special fixture, as shown in Figure 3, was applied. This fixture is similar to the fixture for spot-welded joints proposed by Lee et al. [22]. The detailed loading fixture of the specimen is available in the literature [23]. The loading angle of 0° with cross-shaped specimen is identical to the tensile-shear loading. Thus, the tensile-shear specimen was adopted for evaluating a loading angle of 0°. The SPR joint specimens consisted of magnesium alloy as the top plate and aluminum alloy as the bottom plate. In the tensile test of the AZ31 magnesium alloy, the tensile strengths of the specimens in the rolling direction and in the transverse direction were found to be similar at approximately 227 MPa. On the other hand, the elongation at failure of the specimen in the transverse direction was close to 10%, which is nearly 0.6 times lower than the rolling direction value. The rolling direction of the top magnesium plate was installed parallel to the loading direction during specimen preparation. The rivets used to join the specimens were supplied from Henrob Ltd. (model number C50541, Herford, German), and they consisted of 0.35% carbon steel. The shank diameter and length of the rivets are 5.3 mm and 5.0 mm, respectively. The riveting force was determined to be 32 kN according to monotonic test results of the tensile-shear specimen geometry produced. To evaluate the static and fatigue strength levels of the SPR joints, a servo-hydraulic universal testing machine (Instron model 8516, Warren, MI, USA) with a capacity of 100 kN was used. Fatigue tests were conducted at a load ratio R = ($P_{min}$/$P_{max}$) of 0.1 at a frequency of 2 Hz to 5 Hz.

**Table 1.** Mechanical properties of the AZ31 and Al5052 sheets.

| Material | σu (MPa) | σy (MPa) | Elong. (%) |
| --- | --- | --- | --- |
| AZ31 | 277.1 | 147.8 | 10 |
| Al5052 | 234.4 | 161.8 | 9 |

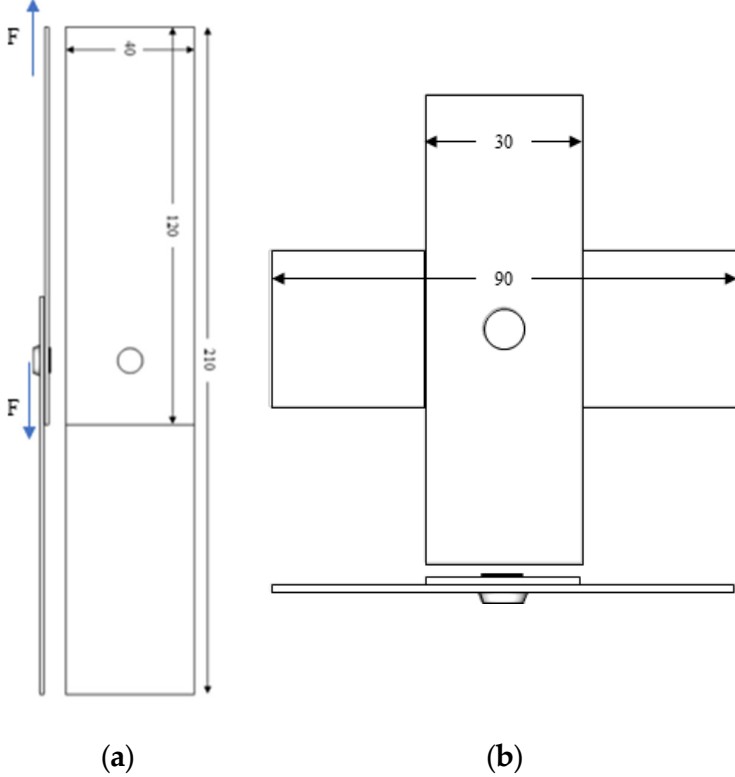

**(a)** **(b)**

**Figure 2.** Geometries and dimensions of the (**a**) tensile-shear and (**b**) cross-shaped specimens (mm).

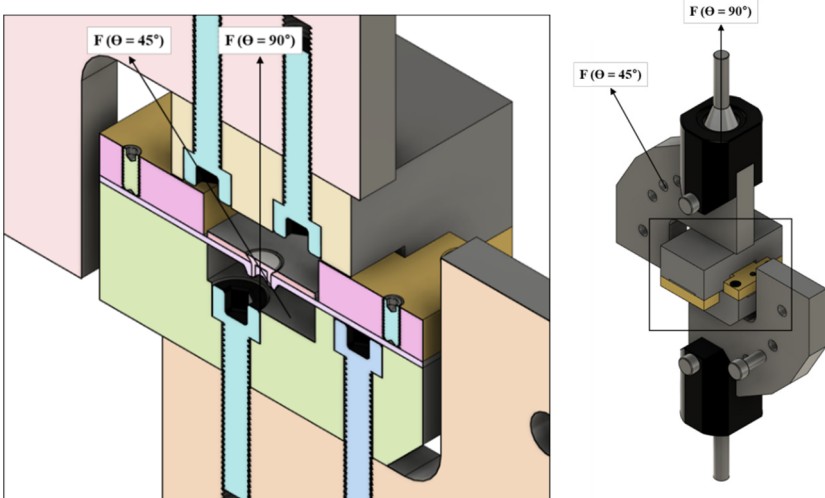

**Figure 3.** Loading angle fixture for the cross-shaped SPR joint specimen.

*2.2. SPR Joining Analysis*

A two-dimensional axisymmetric FEA was conducted for the SPR joining process with ABAQUS/explicit as the solver and HyperMesh as the pre- and post-processor. The components used for the SPR joints were composed of punches, blank holders, dies, rivets, a top plate, and a bottom plate, as shown in Figure 4. A single joint of top magnesium and bottom aluminum plates 1.5 mm thick was modelled. Only half of the specimen was modelled due to symmetry. The blank holder acts as a fixation device for the plates on the die. The punch pushes the rivet in the z direction in the up and down directions to join the plates. The die, punch, and blank holder were modelled as 2D rigid elements (R3D4) and the rivets and plates were modelled as deformable parts with 3D Hexa elements (C3D8R).

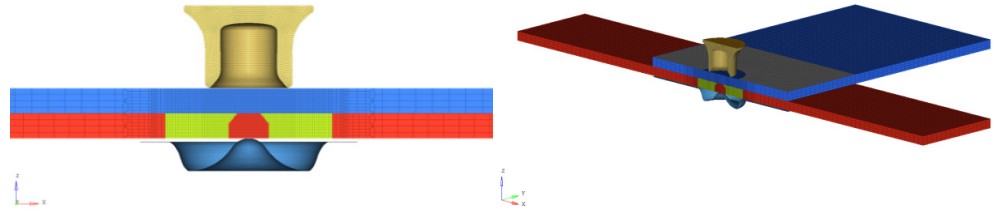

**Figure 4.** Half-section view of the FEA model for SPR joining.

To determine the residual stress distribution in the SPR joint, the simulation was performed in two stages using ABAQUS explicit. The first stage is the riveting simulation, including clamping, piercing, flaring and compression. For the first stage, displacement is prescribed for the blank holder that perpendicularly clamps the top and bottom plates to the die. Then, additional displacement is prescribed for the punch that pushes the rivet through the top plate; the rivet tail flares inside the bottom plate and follows the contours of the die. Finally, the punch stops when it reaches the predetermined force of 32 kN. After the riveting simulation, the second stage is spring-back analysis performed on the final configuration of the joint in order to simulate the release of the punching force. The spring-back was conducted by releasing the punch, blank-holder and die. Stresses remaining inside the elements were considered as residual stress.

For the plate material relationships between the stress and strain with large deformation, the following Johnson–Cook model [24] was adopted.

$$\sigma = [A + B\varepsilon^n][1 + C\ln(\dot{\varepsilon}^*)][1 - T^{*m}] \tag{1}$$

Here, σ is the equivalent stress and ε is the equivalent plastic strain. *A*, *B*, *C*, *m* and *n* are material constants. *A* is the yield stress of the material under the reference conditions, *B* is the strain-hardening constant, *C* is the strengthening coefficient of the strain rate, m is the thermal softening coefficient, and n is the strain-hardening coefficient [25]. The constant $\dot{\varepsilon}^{*}$ and $T^{*m}$ are related to the strain rate and temperature. The three terms in parenthesis in Equation (1) represent the strain-hardening effect, the strain-rate-strengthening effect, and the temperature effect. The values of *C* and *m* were defined as zero because the SPR process is conducted at a constant speed and temperature. This study determined *A*, *B*, and *n* for Johnson–Cook models for Al5052 and AZ31 through curve fitting. The stress–strain curve data for AZ31 and Al5052 alloys used in this analysis were sourced from the literature [9,26] in order to define the Johnson–cook deformation model. Table 2 summarizes the values of *A*, *B*, and *n* determined in this study. Figure 5 shows the stress–strain relationship curve of the Al5052 and AZ31 alloys adopting the values of *A*, *B* and *n*, confirming that these property values are appropriate to describe the stress–strain curves of the two materials.

**Table 2.** Johnson–cook model parameters.

|  | *A* (MPa) | *B* (MPa) | *n* |
|---|---|---|---|
| AZ31 | 147.8 | 282.9 | 0.3 |
| Al5052 | 161.8 | 366.1 | 0.63 |

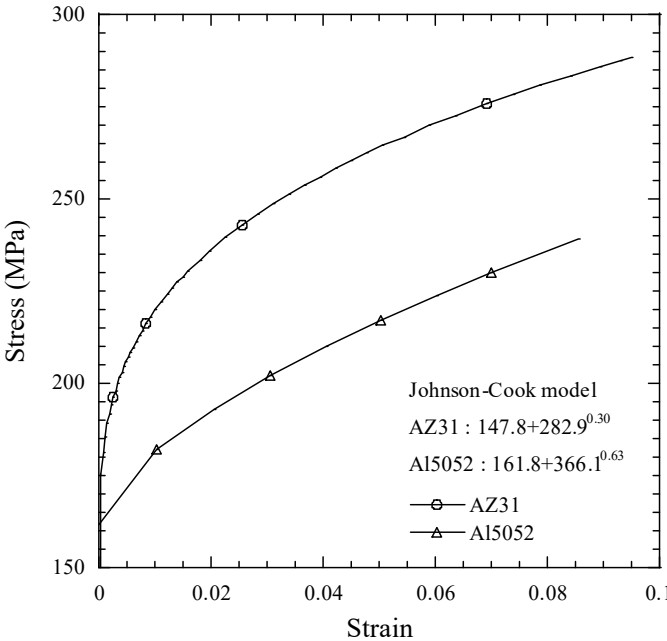

**Figure 5.** Stress-strain curves of AZ31 and Al5052 alloys with the Johnson–cook deformation model.

In order to allow the rivet to pierce through the top plate, elements were deleted according to the Johnson–Cook damage criterion [27]. The Johnson–Cook model can be written as follows:

$$\varepsilon_f = [d_1 + d_2 \exp(d_3 \frac{\sigma_m}{\sigma_{eq}})][1 + d_4 \ln(\dot{\varepsilon}_p^{*})](1 + d_5 T^{*}) \tag{2}$$

In this equation, $d_1$ to $d_5$ are the damage model constants, $\dot{\varepsilon}_p^{*}$ is the non-dimensional strain rate, $\sigma_m$ is the mean stress, $\sigma_{eq}$ is the equivalent stress and $T^{*}$ is the non-dimensional temperature. The parameter $d_1$ is a constant, while the parameters $d_2$ and $d_3$ are related to stress triaxiality, allowing for the deletion of elements at different strain thresholds depending on the stress state. The parameter $d_4$ is related to the strain rate. The parameter

$d_5$ is related to the temperature, and it was set to zero in this study because the SPR process is conducted at an ambient temperature. Damage parameters in the model for the Al5052 and AZ31 alloys were sourced from the literature [28,29]. Table 3 summarizes the values of the damage model constants $d_1$ to $d_4$ determined in this study. The following values for the friction coefficient µ between the die and different parts in the model were set as follows: $\mu = 0.19$ for the aluminum to magnesium plates, $\mu = 0.3$ for the bottom aluminum plate to the die, and $\mu = 0.3$ for the top magnesium plate to the blank holder.

**Table 3.** Johnson–cook damage parameters for the AZ31 and Al5052 alloys [28,29].

| J-C Parameters | $d_1$ | $d_2$ | $d_3$ | $d_4$ |
|---|---|---|---|---|
| AZ31 | 0.35 | 0.6 | 0.45 | 0.21 |
| Al5052 | 0.31 | 0.04 | 1.72 | 0.01 |

### 2.3. Structural Analysis of an SPR Joint

The structural analysis was conducted on two types of cross-section profiles of the SPR joints. The first is the cross-section profile of a joint produced with a punching force of 32 kN, as acquired from a SEM (scanning electron microscopy) observation, as shown in Figure 6. The second is the cross-section profile formed after the FEA analysis for SPR joining. After performing the SPR joining analysis using ABAQUS, the cross-section profile of the SPR joint was exported using HyperView (version 7.0, Altair, Troy, MI, USA) and modelled, as shown in Figure 7. Comparing the cross-section profile of the SPR joint model after the FEA joining analysis, as shown Figure 7, to the actual cross-sectional shape in Figure 6, we identified similarities. The structural analysis model for the cross-shaped specimen with an SPR joint is shown in Figure 8. This model consists of a 3D element (C3D8R) with 209,452 elements and 223,521 nodes. The friction values between the plate and the plate and the plate and the rivet were determined using the contact pair option, and the coefficient of friction between AZ31 and Al5052 was set to 0.19. The top plate is fully fixed so that the boundary condition is identical to that in the actual fatigue experiment. The analysis was performed with an applied load in the directions of 45° and 90°. For a load angle of zero, a structural analysis was performed by modelling of the tensile-shear specimen feature. In order to estimate the effect of the residual stress on the fatigue lifetimes of the SPR joint, a structural analysis of the cross-section formed after the FEA SPR joining analysis was conducted with and without the residual stress distributions.

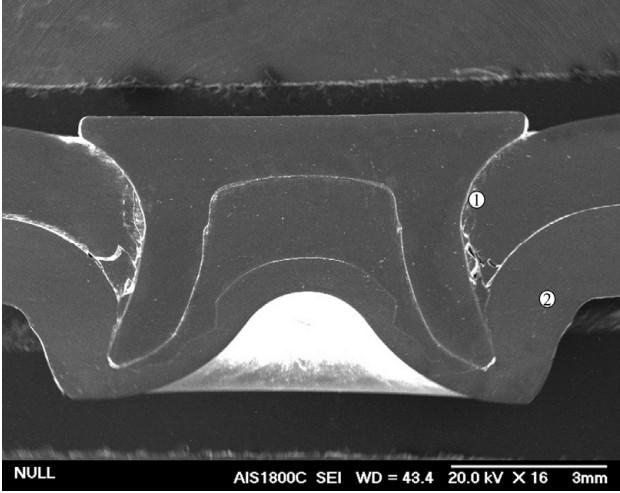

**Figure 6.** Observed the cross-section of the SPR joint by SEM and measured points of the residual stress.

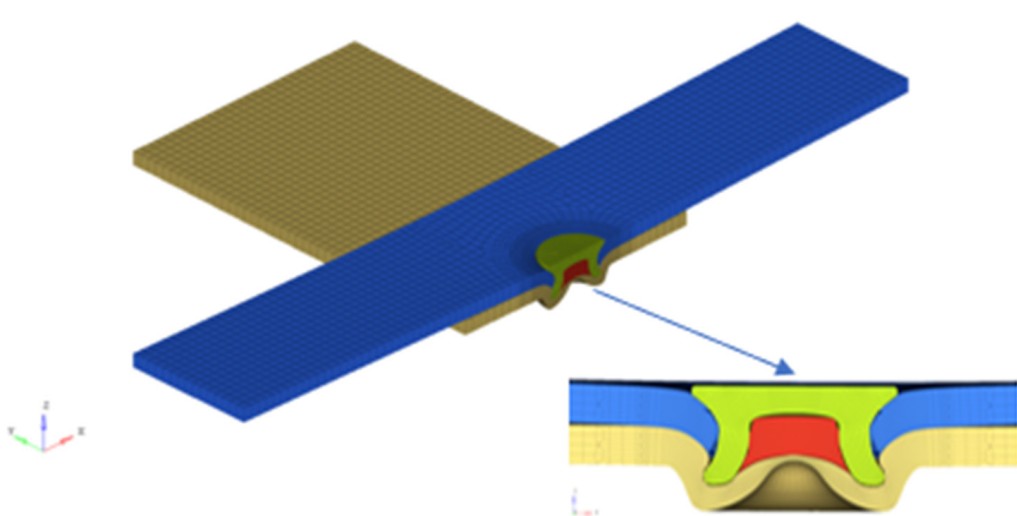

**Figure 7.** Overview of FE models and a detail SPR joint for the cross-shaped specimen.

### 2.4. X-ray Diffraction Measurements

In order to measure the residual stresses on the surface of the SPR joint, X-ray diffraction measurements were taken using an X-ray diffraction analyzer (Bruker Co. U.K. D8 Advance). Additionally, an analysis was done using Bragg's law to calculate the lattice spacing. The {hkl} plane used for the diffraction of magnesium was {110}, with the 2θ angle being approximately 57.4°. The {hkl} plane used for the diffraction of aluminum was {311}, with the 2θ angle being approximately 78.2°. The measurements were taken from the locations '1' on the top magnesium alloy and '2' on the bottom aluminum plates of the cross-section of the SPR joint in Figure 6. The strains were measured along the laboratory coordinated axis, perpendicular to the particular {hkl} plane used in the diffraction. The 'rocking technique" was used to calculate the residual stress. The detailed residual calculation by the "rocking technique" is available in the literature [30].

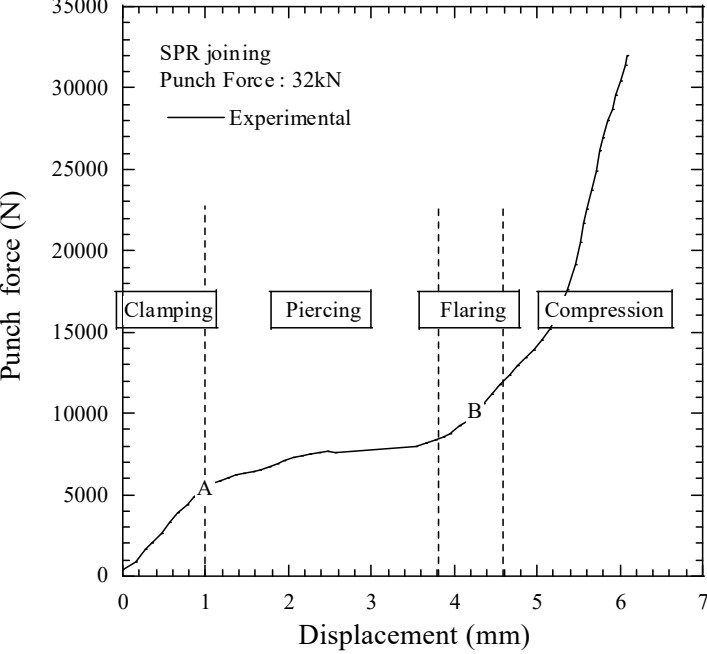

**Figure 8.** *Cont.*

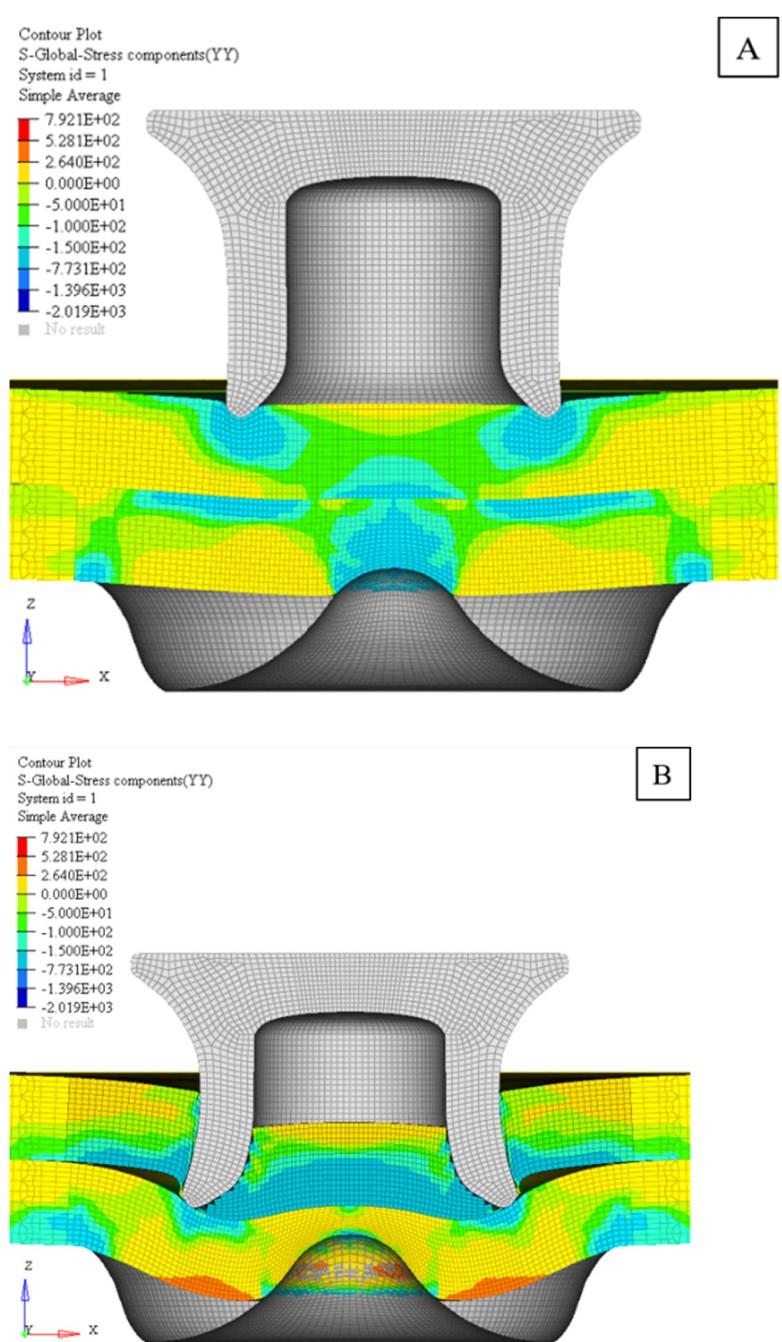

**Figure 8.** The punch force-displacement curve and stress distribution at the stages (**A**,**B**) during the SPR process.

## 3. Results

### 3.1. Residual Stress Distribution of an SPR Joint through the FEA Joining Analysis

Figure 8 shows the experimental punch force–deformation curve and stress distribution at two selected points A and B during the riveting process, where points A and B correspond to the onset of the piercing process and the middle of the flaring process, respectively. As shown in Figure 8, the stress contour corresponding to the onset of the piercing process (A point) exhibits compressive stresses on the top plate in contact with the tip of the rivet tail and the bottom part of the bottom plate in contact with the die center. In the middle of the flaring process (B point), compressive stresses are concentrated on the overall bottom part of top plate in contact with the circumference of the rivet shank and tensile stresses exist on the overall top center part of the bottom plate.

When the punch load was removed, additional deformation occurred on the joint due to spring-back. After the spring-back process, residual stresses were present in the SPR joint. The residual stress distribution in terms of the radial stress $\sigma_r$ and hoop stress $\sigma_\theta$ of the sheets after spring-back is shown in Figure 9a,b, respectively. As shown in Figure 9a, the residual stresses in the radial direction $\sigma_r$ in the top and bottom plates are negative near the rivet hole. The magnitude of the radial compressive is highest at the top center of the bottom plate. Meanwhile, the magnitude of the radial tensile stress is largest at the button of the bottom plate, close to the rivet tail. Figure 9b shows the residual stresses in the hoop direction $\sigma_\theta$. Higher compressive residual stresses are distributed in the top plate near the rivet hole compared to the radial residual stress. The magnitude of tensile hoop stress is largest at the button of the bottom plate, close to the rivet tail. This location is similar to the radial stress case. The radial and hoop stresses may be induced by residual clamping force from the interlock of the rivet and the constraint of plastic expansion within the sheets.

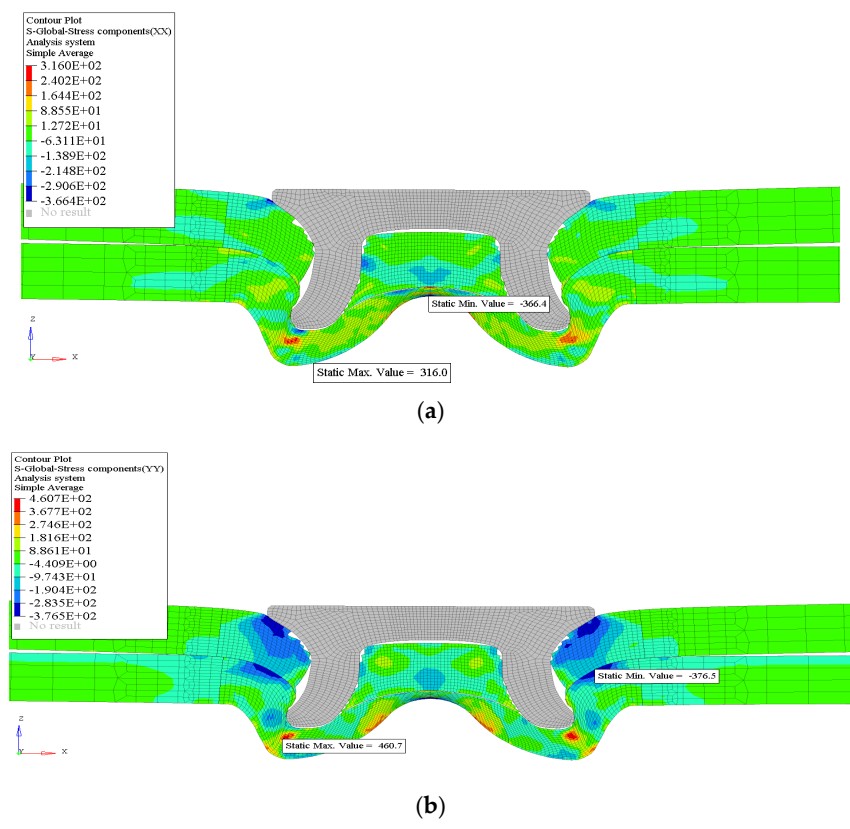

**(a)**

**(b)**

**Figure 9.** Residual stress (**a**) $\sigma_r$ and (**b**) $\sigma_\theta$ distributions for the SPR joint after spring-back.

### 3.2. Experimentally Measured Residual Stresses of an SPR Joint

The residual stresses at locations '1' and '2' in Figure 6 were measured using the X-ray diffraction method. The stress ($\sigma_{yy}$) normal to the X-Z plane of the rivet joint in Figure 9b corresponds to the measured residual stress ($\sigma_\theta$). Thus, the measured residual stresses ($\sigma_\theta$) were compared to the stress ($\sigma_{yy}$) determined through the FEA simulation in order to verify the validity of the distribution of residual stresses obtained from the FEA joining analysis. The measured and simulated residual stresses ($\sigma_\theta$) of location '1' on the top magnesium plate were identified as $-110.3 \pm 23.4$ MPa and $-126.5$ MPa, respectively. The measured and simulated residual stresses ($\sigma_\theta$) of location '2' on the bottom aluminum plate were identified as $-126.6 \pm 28.5$ MPa and $-152.9$ MPa, respectively. These results partially confirm that the FEA method can appropriately determine the residual stress distribution of an SPR joint within an error of 17%, even with only two-point data.

### 3.3. Evaluation of Fatigue Strength of an SPR Joint

Static tests were conducted on the tensile-shear and the cross-shaped specimens at loading angles of 45° and 90°. Figure 10 shows the applied load against the displacement for the specimens. The peak load of the tensile-shear specimen was 3316 N. And, the peak loads of the cross-shaped specimens at the loading angles of 45° and 90° were 1955 N and 1644 N, respectively, suggesting that a decrease in the monotonic strength as the load angle is increased. This behavior is consistent with the general behavior of the strength of a spot joint, such as a spot-welded joint or a clinch joint [31]. As the loading angle increases, shear force with respect to the rivet component decreases and the tensile force component increases. Thus, the separation of the rivet roots from the bottom plate due to bending moment by tensile force is more critical on the specimen compared to the separation of the rivet roots from the bottom plate due to shear force.

The fatigue test results of the specimens at loading angles of 0°, 45°, and 90° are summarized in Table 4. Figure 11 is a plot of the fatigue lifetime as a function of the load amplitude at loading angles of 0°, 45°, and 90°. The relationships between the number of cycles ($N_f$) and the load amplitude at the loading angles of 0°, 45°, and 90° are $P_{amp} = 4696.0 N_f^{-0.12}$, $P_{amp} = 1852.4 N_f^{-0.15}$, and $P_{amp} = 2795.9 N_f^{-0.21}$, respectively. The load amplitudes corresponding to the fatigue limit, based on a $10^6$ cycles lifetime, were 895 N, 233 N, and 154 N for the loading angles of 0°, 45° and 90°, respectively. This fatigue limit corresponds to 27% of the static strength ($P = 3316$ N) for the loading angle of 0° with tensile-shear loading. When the loading angle was 45°, the fatigue limit was found to be approximately 12% of the static strength ($P = 1955$ N), and, when the loading angle is 90°, the load amplitude (fatigue limit) is approximately 9% of the static strength ($P = 1644$ N).

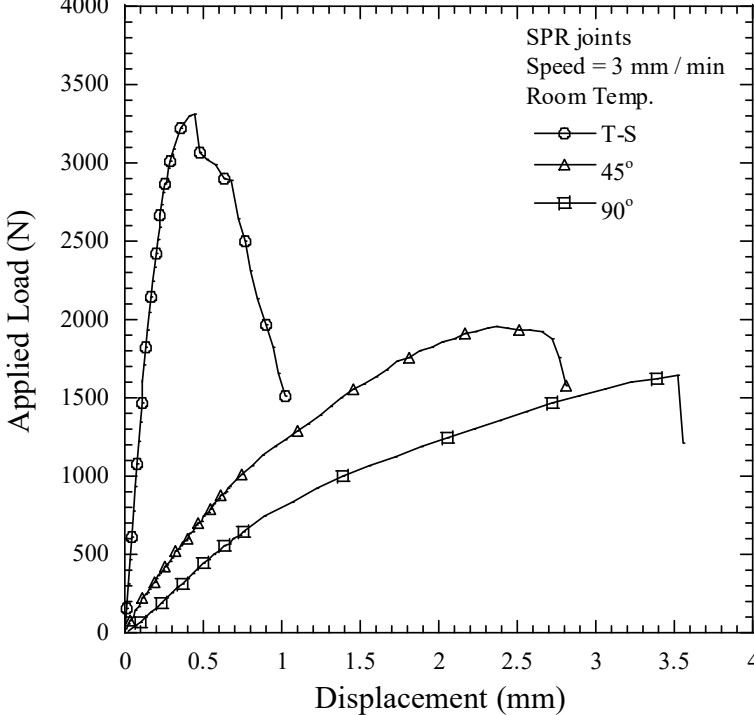

**Figure 10.** Comparison of the static load versus the displacement curves of SPR specimens at different loading angles.

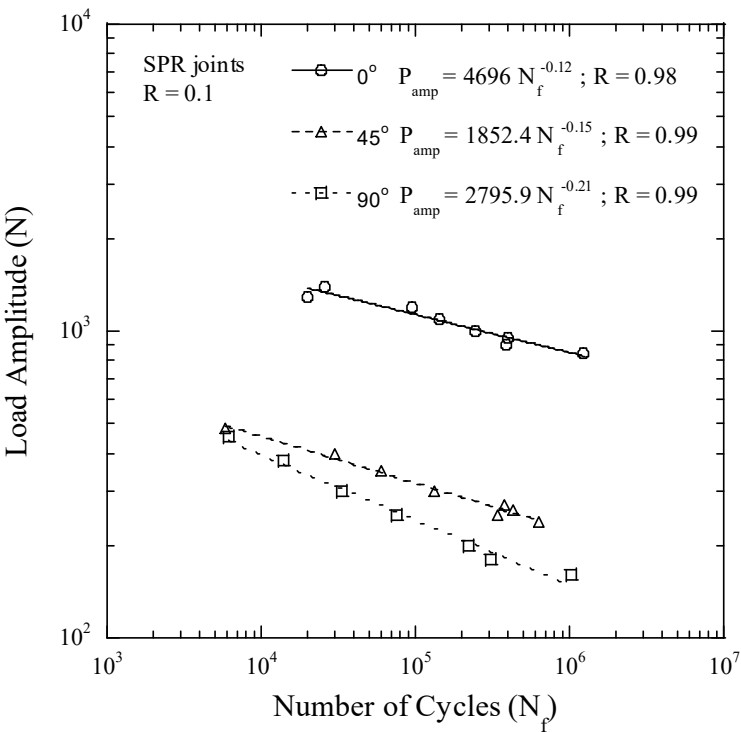

**Figure 11.** Comparison of the load amplitude against the number of cycle plots for SPR joints at different loading angles.

**Table 4.** Experimental fatigue lifetime data of SPR joints under various loading angles.

| Loading Angle | $P_{amp}$ (N) | $N_f$ (Cycles) | Loading Angle | $P_{amp}$ (N) | $N_f$ (Cycles) |
|---|---|---|---|---|---|
| 0° | 1400 | 25,858 | 45° | 270 | 375,720 |
| 0° | 1300 | 19,844 | 45° | 260 | 435,110 |
| 0° | 1200 | 95,389 | 45° | 250 | 343,070 |
| 0° | 1100 | 143,354 | 45° | 240 | 625,150 |
| 0° | 1000 | 244,825 | 90° | 450 | 6148 |
| 0° | 950 | 398,884 | 90° | 380 | 13,875 |
| 0° | 900 | 385,692 | 90° | 300 | 33,481 |
| 0° | 850 | 1,226,220 | 90° | 250 | 74,910 |
| 45° | 480 | 5916 | 90° | 200 | 224,111 |
| 45° | 400 | 29,905 | 90° | 180 | 305,793 |
| 45° | 350 | 59,430 | 90° | 160 | 1,030,904 |
| 45° | 300 | 133,120 | - | - | - |

### 3.4. Fatigue Life Estimation

It has been shown that the parameter of the load amplitude is not appropriate for determining correlations between the fatigue strength of SPR joints at various load angles, as shown in Figure 11. Therefore, fatigue lifetimes are evaluated by adopting parameters such as von-Mises stress, maximum principal stress, and SWT fatigue parameter [32]. For this evaluation, these parameters were assessed by applying the FEA model for the

specimen with the cross-section of the SPR joint that was observed by SEM, as shown in Figure 6. The SWT fatigue parameter [32] is expressed as shown below,

$$\sigma_1^{\max}\frac{\Delta\varepsilon_1}{2} = f(N_f) \tag{3}$$

where $\Delta\varepsilon_1/2$ is the maximum principal strain amplitude and $\sigma_1^{\max}$ is the maximum stress on the $\Delta\varepsilon_1$ plane. Figure 12a–c plot the fatigue lifetime of an SPR joint under various loading angles as a function of the effective stress, maximum principal stress, and SWT fatigue parameter, respectively. As shown in Figure 12, the effective stress, maximum principal stress, and SWT fatigue parameter are not appropriate to correlate the fatigue lifetimes for SPR joints under various loading conditions.

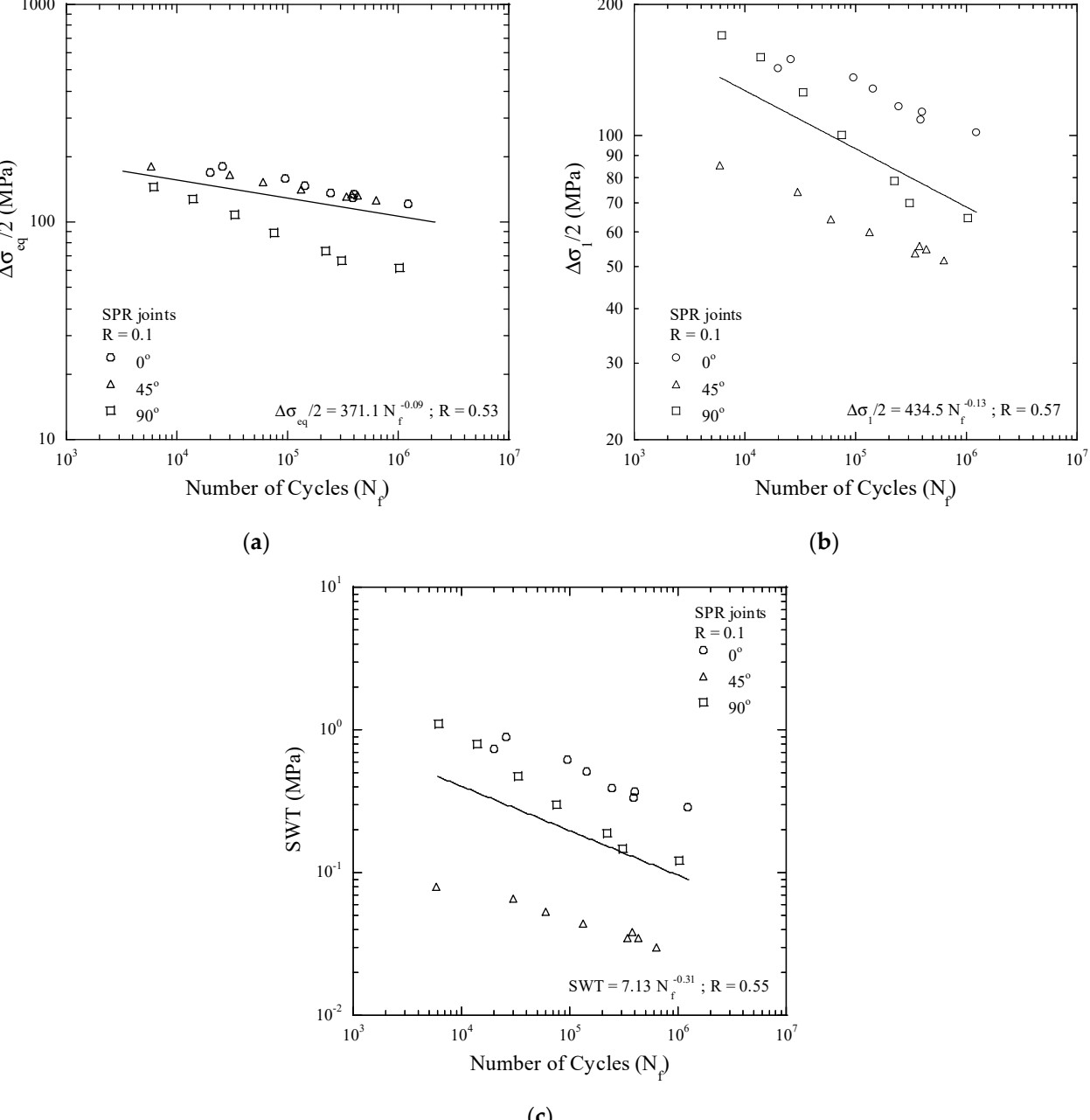

**Figure 12.** Experimental fatigue lifetimes of the SPR joint specimens under various loading conditions as a function of (**a**) the von-Mises stress, (**b**) the maximum principal stress, and (**c**) the SWT fatigue parameter.

The fatigue lifetime of the cross-shaped specimen was evaluated by applying the equivalent stress intensity factor of the spot-welded joint specimen proposed by Zhang [33]. Zhang defines the equivalent stress intensity factor for a spot-weld specimen using Equation (4).

$$K_{eq} = \frac{0.637F}{d\sqrt{t}} \sqrt{\cos^2\theta + \frac{\zeta^2(b-d)^2}{12t^2}\sin^2\theta} \tag{4}$$

Here, F denotes the applied force and $\theta$ is the angle of the applied force. In addition, $d$, $t$ and $b$ are the rivet diameter (=5.3 mm), the thickness of the specimen (=1.5 mm), and the width of the specimen (=30 mm). $\zeta$ is the correction factor of the specimen geometry [23]. The correction factor $\zeta$ for the current cross-shaped specimen was determined to be 1.19. At a load angle of 90° for the cross-shaped specimen, the equivalent stress intensity factor becomes $K_{eq} = \frac{0.637F}{d\sqrt{t}}\sqrt{\frac{\zeta^2(b-d)^2}{12t^2}}$. Meanwhile, for a loading angle of 45° for the cross-shaped specimen, the specimen is subjected to both tensile and shear forces. In other words, because the loading angle is 45°, the tensile and shear forces acting on the specimen are Fsin45° and Fcos45°, respectively, which are both 0.707. Therefore, the equivalent stress intensity factor for a loading angle of 45° is defined by Equation (5),

$$K_{eq} = \sqrt{0.707K_0^2 + 0.707K_{90}^2} \tag{5}$$

where $K_0$ and $K_{90}$ are the equivalent stress intensity factors at the loading angles of 0° and 90°, respectively. Meanwhile, the equivalent stress intensity factor of the tensile-shear specimen corresponding to a load angle of 0° becomes $K_{eq} = 0.637F/d\sqrt{t}$.

Figure 13 presents the results of the assessment of the fatigue lifetimes at three loading angles by applying the equivalent stress intensity factor. As shown in Figure 13, the equivalent stress intensity factor ($R \approx 0.86$) shows a good correlation with the experimental fatigue lifetime data compared to the plots based on the effective stress, maximum principal stress, and SWT fatigue parameter. The relationship between the equivalent stress intensity factor amplitude ($\Delta K_{eq}/2$) and the number of cycles ($N_f$) was found to be $\Delta K_{eq}/2 = 31.0N_f^{-0.17}$.

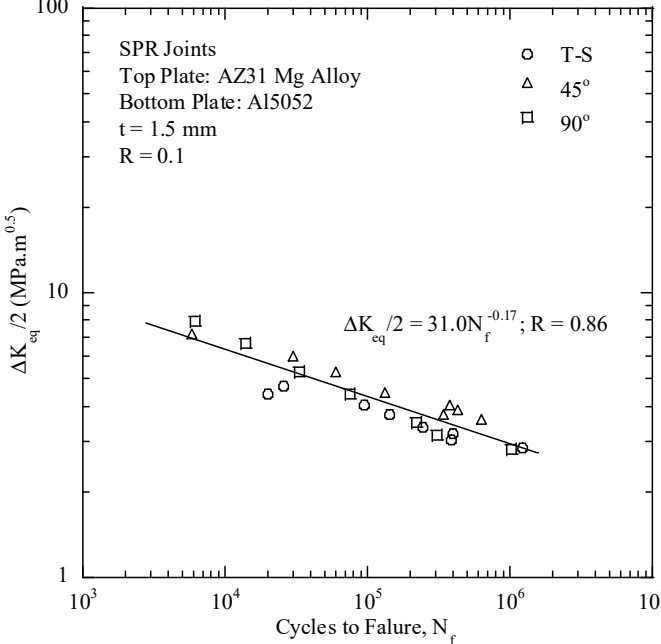

**Figure 13.** Experimental fatigue lifetimes of the SPR joint specimens under various loading conditions as a function of the equivalent stress intensity factor amplitude.

The equivalent stress intensity factor is found to be the most appropriate parameter to correlate the fatigue lifetimes of the current Mg/Al SPR joints and Al/Al SPR joints [23,26] under various loading angles. In addition, the maximum principal stress was reported to be the most appropriate to correlate the fatigue lifetimes of the Mg/Steel SPR joints [9]. However, it is not currently clear why the equivalent stress intensity factor and maximum principal stress appropriately predict the fatigue lifetimes of the Mg/Al and Mg/Steel SPR joints under various loading angles, respectively. This requires more in-depth investigation.

### 3.5. Effect of Residual Stresses on the Fatigue Lifetimes of SPR Joints

To evaluate the effect of residual stresses on the fatigue lifetimes of SPR joints, the fatigue strength was estimated while considering and not considering the residual stresses. It is difficult to identify the detailed residual stress distribution of SPR joints when using experimental fatigue tests. Therefore, a structural analysis was conducted on the cross-section profile of an SPR joint obtained through the SPR joining analysis with and without the residual stress distribution. After obtaining the structural analysis results, the fatigue lifetimes were assessed by the parameter of the maximum principal stress of the fatigue specimens. Figure 14 shows a comparison of the fatigue lifetimes of SPR joint profile models after the joining analyses with and without the residual stresses as a function of the maximum principal stress. The stress amplitudes at a $10^6$ cycles lifetime with and without the residual stress were 78.4 MPa and 90.2 MPa, respectively. This suggests that the compressive residual stresses of the joint reduce the stress amplitude by 13% at a $10^6$ cycles lifetime.

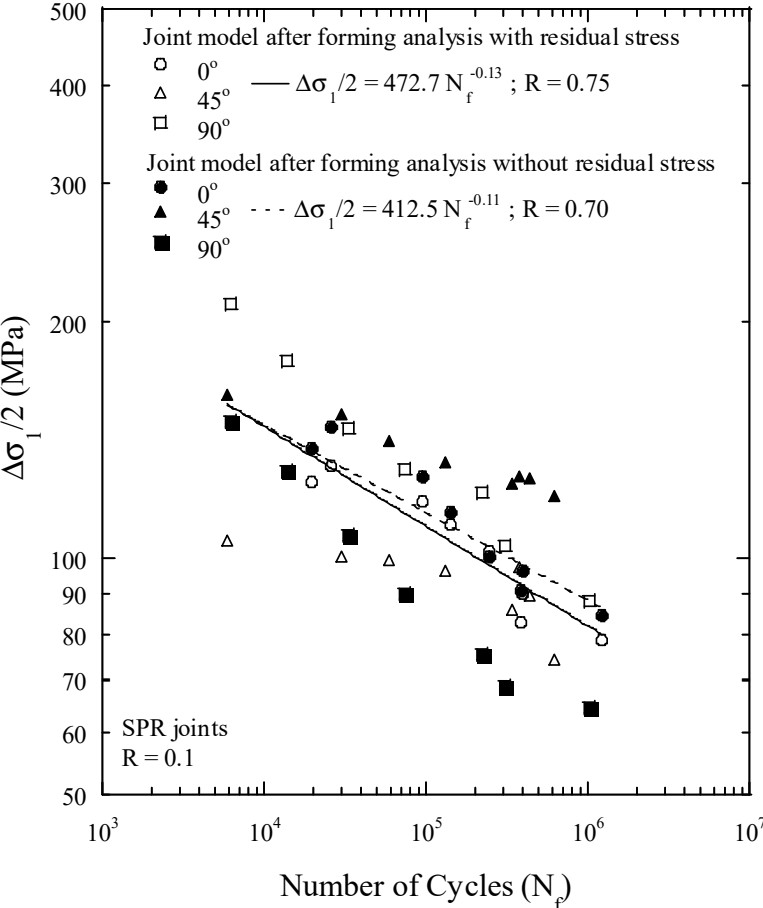

**Figure 14.** Comparison of fatigue lifetimes of SPR joint profile models according to a joining analysis with and without residual stress as a function of the maximum principal stress.

The fatigue lifetime of a component is made up of initiation and propagation periods. At a shorter fatigue lifetime, where the crack propagation process dominates, the residual stress has less of an effect. However, in the high-cycle fatigue regime, where crack initiation dominates, the effects of residual stress can be significant [34]. Figure 15 shows schematics of the effect of residual stress on the fatigue lifetime with and without residual stress for an SPR joint. For the fatigue lifetimes of SPR joints without residual stresses, represented by the dashed line in Figure 15, a reduction of the stress amplitude by 13% at a fatigue lifetime of $10^6$ cycles arises if residual stress is added to the analysis of the joint. In other words, if the amount of the stress amplitude (=11.8 MPa) is lowered from 90.2 MPa to 78.4 MPa at a fatigue lifetime of $10^6$ cycles, fatigue lifetime can be expected to increase by approximately 3.4 million cycles from $10^6$ cycles. At a fatigue lifetime of $10^4$ cycles, as shown in Figure 15, for an SPR joint without residual stress, as represented by the dashed line in Figure 15, the stress amplitude of 149.8 MPa will be lowered to 142.8 MPa if residual stress is present in the joint. This is expected to exhibit a fatigue lifetime of 17,000 cycles, representing an increase of approximately 70%. As shown here, in the low-cycle fatigue regime, there is less of an effect of residual stress on extending the fatigue lifetime, compared to the high-cycle life case. In the low-cycle fatigue regime, the crack initiation period is shorter than in the high-cycle fatigue regime. As a nucleated crack propagates, the residual stress in front of the crack tip is more released, resulting in a smaller effect of residual stress. However, in the high-cycle fatigue regime, the crack initiation period is longer than the crack propagation period. The compressive residual stress suppresses crack initiation, having the effect of increasing the crack initiation portion and the total fatigue life.

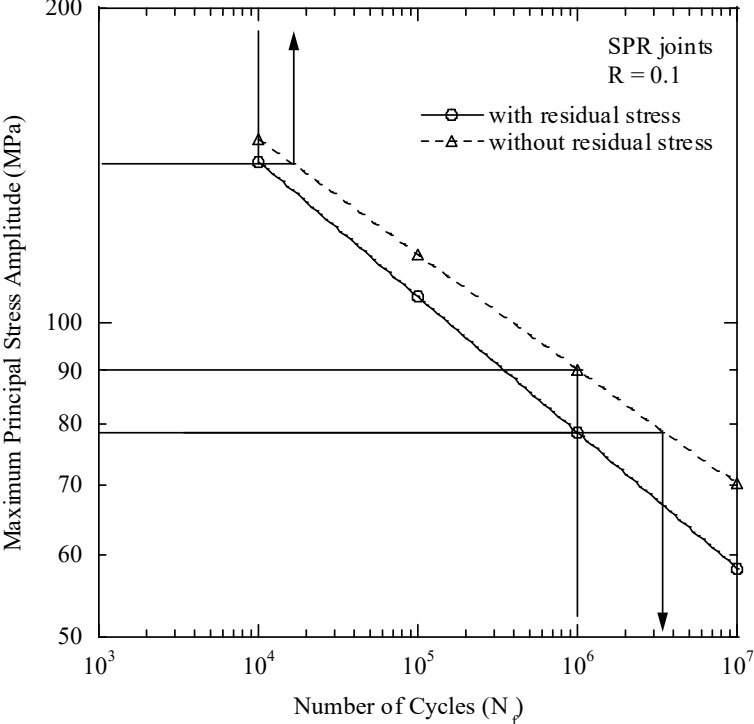

**Figure 15.** Effects of compressive residual stress on the fatigue lifetime of an SPR joint.

## 4. Discussion

It is meaningful to compare the fatigue strengths of SPR joints of dissimilar magnesium alloy plates. When complex loads act on an SPR joint intended to join dissimilar material plates, it is necessary to utilize a plate material with high strength for the bottom plate where higher stresses are distributed, rather than on the top plate. In addition, more brittle material plates tend to be placed on the top plate. A plate consisting of a brittle material complicates the forming of a button with the die profile shape due to its lower

formability during SPR joining if such a plate is utilized as the bottom plate. If dissimilar magnesium plates, such as magnesium alloy and steel or magnesium alloy and aluminum alloy combinations, are joined by the SPR technique, the magnesium plate should be utilized as the top plate for both cases due to its lower strength and ductility. In other words, it is inevitable for the magnesium alloy plate to be utilized as the top plate in the event of SPR joining with dissimilar magnesium plate combinations.

Figure 16 shows the fatigue lifetimes of two types of SPR joint specimens as a function of the equivalent stress intensity factor. One type is the combination used in the current experiments involving magnesium AZ31 alloy as the top plate and aluminum Al5052 alloy as the bottom plate (designated T.M-B.A). The other type is a combination of magnesium AZ31 alloy as the top plate and cold-rolled mild steel as the bottom plate (designated T.M-B.S) [9]. For the T.M-B.S specimens, the relationship between the stress intensity factor amplitude and the number of cycles is $\Delta K_{eq}/2 = 18.3 N_f^{-0.13}$ [9]. Figure 16 shows that the T.M-B.A specimen has overall higher fatigue strength than the T.M-B.S specimen under the present testing condition. In addition, the fatigue strengths of the two types of specimens are similar as the fatigue lifetime approaches a high number of cycles of $10^6$. Note that the static strengths of the T.M-B.A and T.M-B.S specimens produced with the identical die and rivets were found to be 3315 N and 3771 N [9], respectively, suggesting that the monotonic T.M-B.S specimen exhibits approximately 14% higher strength than the T.M-B.A specimen. Therefore, it can be expected that a combination that utilizes steel with higher strength as the bottom plate, where higher stress is concentrated, will exhibit higher fatigue strength and longer fatigue lifetimes as opposed to a combination that utilizes aluminum with lower strength as the bottom plate, with an identical magnesium top plate for both combinations. In contrast, as shown in Figure 16, the fatigue strengths of the T.M-B.S specimens exhibit lower values overall than those of the T.M-B.A specimens under the testing condition. This outcome can be due partially to the magnitude of the difference in the hardness and fretting amplitude of the facing materials.

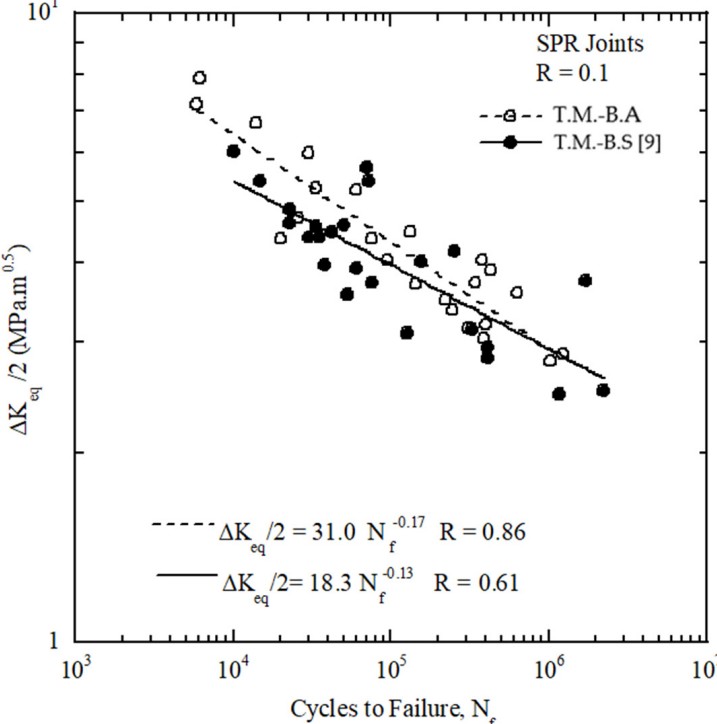

**Figure 16.** Comparison of the fatigue lifetimes of the T.M-B.A and T.M-B.S [9] SPR joints under three loading conditions.

As the SPR joint is a type of mechanical joint, when cyclic loads are applied, the contact surfaces between the two plate materials are subjected to small amplitude motion due to variations in the deformation with differences in the elastic modulus. It was observed that fatigue crack initiation of an SPR joint generally occurred due to fretting wear on the contact points between the two plates or between the rivet and the plates [8,10]. The T.M-B.A combination is judged to have an advantage over the T.M-B.S combination regarding the degree of resistance to fatigue crack initiation. The larger the difference in the hardness of the two facing plate materials, the more cracks easily initiate on the plate material with a lower hardness value. Because the difference in the hardness of the facing plate materials of the T.M-B.S combination is greater than that of T.M-B.A combination, the contact surfaces of the magnesium plate and the steel plate of the T.M-B.S specimen have a shorter crack initiation lifetime than those of the magnesium plate and the aluminum plate of the T.M-B.A specimens. The high-cycle fatigue lifetime of an SPR joint is controlled by the crack initiation resistance rather than by the crack propagation resistance. Therefore, it can be expected that the T.M-B.A specimen may exhibit a longer fatigue lifetime than the T.M-B.S specimen.

Another factor is the fretting amplitude of the facing plates of SPR joints. For the T.M-B.S joint, the fretting amplitude of the facing material contacts is larger compared to the T.M-B.A joint; under the same load, the relative displacement of the contact point between the top and bottom plates is proportional to the difference in elastic modulus of the two materials. The elastic modulus for AZ31 alloys, Al5052 aluminum alloys and steel are approximately 45 GPa, 70 GPa and 210 GPa, respectively. Therefore, it is deemed that T.M-B.S, which has a larger relative contact displacement that causes the fretting wear under cyclic loading, is more susceptible to crack initiation compared to the T.M-B.A combination.

The fatigue lifetimes of SPR joints with dissimilar plate materials in the high-cycle fatigue regime are governed by the fatigue crack initiation resistance of the plate with the lower strength level. Therefore, in a combination of magnesium alloy and steel or magnesium alloy and aluminum alloy, it is deemed that the fatigue crack initiation resistance of the top magnesium plate of these joints governs the high-cycle fatigue lifetimes of the joints, as magnesium alloys generally exhibit lower strength than steels or aluminum alloys. Figures 17 and 18 show the results of comparisons of the calculated and experimental observed fatigue lifetimes of these two specimen combinations by adopting the equivalent stress intensity factor amplitude relationship of T.M-B.A specimens and T.M-B.S specimens [8], respectively. The two dashed lines, above and/or below the diagonal and parallel to it, correspond to the error of the fatigue lifetime within a factor of three. Therefore, the fatigue strengths of SPR joints of dissimilar magnesium plates under various loading conditions can be adequately predicted by the equivalent stress intensity factor amplitude correlation for T.M-B.S [9] or T.M-B.A joints regardless of the bottom plate material. This fact again proves that the fatigue lifetimes of the T.M-B.A and T.M-B.S combination specimens were governed by the fatigue resistance of the weaker top magnesium AZ31 plate, which is a common plate material for these two types of specimens. For the SPR joints combined with a magnesium alloy plate, the magnesium plate is almost impossible to apply to the bottom plate due to its low ductility. Therefore, regarding the fatigue design of SPR joints combined with magnesium plates and other material plates, it is considered that the fatigue strengths of these joints will be nearly identical regardless of the material of the bottom plate, although the static strength depends on the strength of the bottom plate.

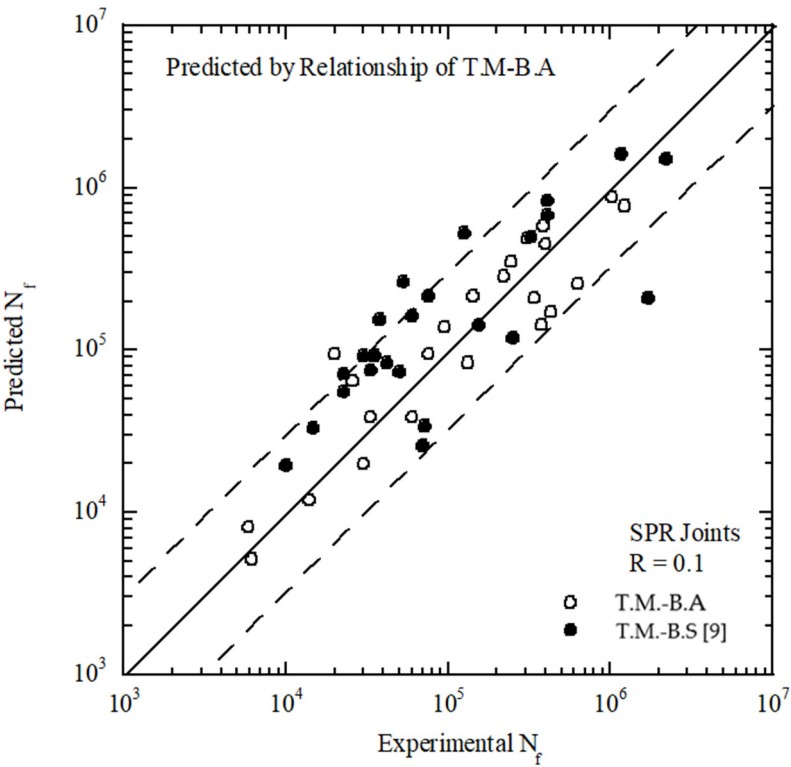

**Figure 17.** Fatigue lifetime predictions of the T.M-B.A and T.M-B.S [9] SPR joints under three loading conditions adopting the relationship of T.M-B.A.

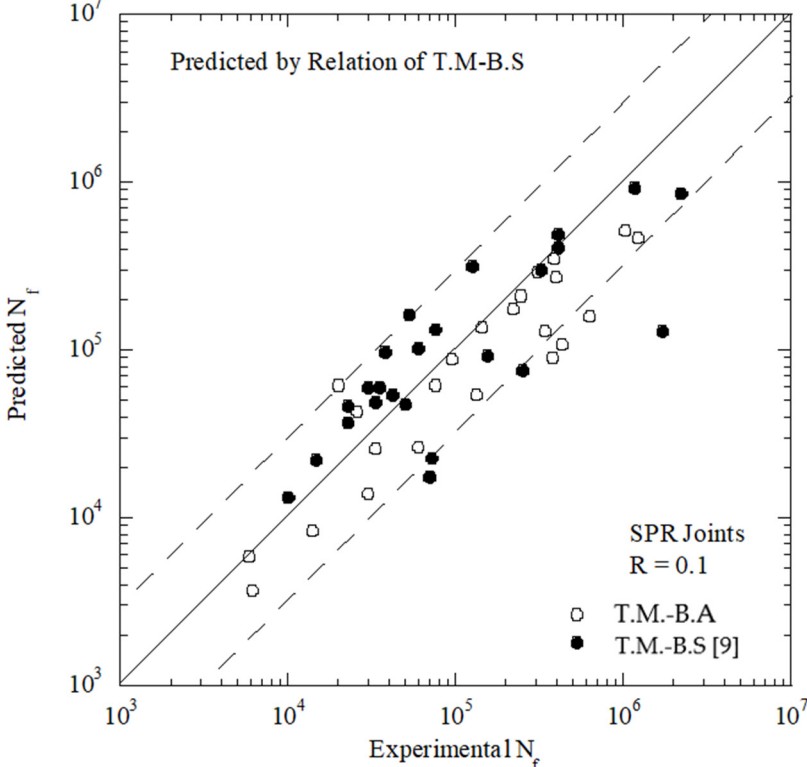

**Figure 18.** Fatigue lifetime predictions of the T.M-B.A and T.M-B.S SPR joints under three loading conditions adopting the relationship of T.M-B.S [9].

## 5. Conclusions

In this study, the residual stresses generated on SPR joints used to join dissimilar materials of magnesium alloy (AZ31) and aluminum alloy (Al5052) were determined by an FEA for SPR joining and experimental measurements. The effects of these residual stresses on the fatigue lifetimes of the joints were quantitatively evaluated. The fatigue lifetimes of SPR joints were predicted by an appropriate fatigue strength parameter at loading angles of $0°$, $45°$ and $90°$. The results are summarized below.

1. Residual stress was measured using an X-ray-diffraction analyzer for two measurement points of SPR joints of the magnesium alloy and aluminum alloy plates. The value of the hoop stress ($\sigma_\theta$) after the joining analysis by the FEA at an identical measured point showed a maximum error of 17.2% in the experimental results.
2. The maximum principal stress amplitudes at $10^6$ cycles lifetime with and without residual stress were 78.4 MPa and 90.2 MPa, respectively, suggesting that the compressive residual stresses of the joint reduce the stress amplitude by 13% at $10^6$ cycles lifetime. At a fatigue lifetime of $10^6$ cycles, fatigue lifetime is expected to increase by approximately 3.4 million cycles due to the reduced stress amplitude.
3. The fatigue lifetimes of the SPR joints are evaluated by applying the von-Mises stress, maximum principal stress, and equivalent stress intensity factors at loading angles of $0°$, $45°$, and $90°$. The equivalent stress intensity factor is found to be most appropriate.
4. Using the relationship of the fatigue lifetime and the equivalent stress intensity factor under various loading conditions of the T.M-B.S and T.M-B.A SPR joint specimens, it was confirmed that the fatigue lifetime of these two types of specimens could be appropriately predicted within a factor of three. This fact suggests that the fatigue resistance of magnesium AZ31 on the top plate leads to extended fatigue lifetimes of these two types of specimens.

**Author Contributions:** Formal analysis, Y.-I.L.; Writing—original draft, H.-K.K. All authors have read and agreed to the published version of the manuscript.

**Funding:** This study was supported by the Research Program funded by the SeoulTech (Seoul National University of Science & Technology).

**Institutional Review Board Statement:** Not applicable.

**Informed Consent Statement:** Not applicable.

**Data Availability Statement:** The data presented in this study are available on request from the corresponding author.

**Conflicts of Interest:** The authors declare no conflict of interest.

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
