# Peer review of "Effects of Residual Stresses on the Fatigue Lifetimes of Self-Piercing Riveted Joints of AZ31 Mg Alloy and Al5052 Al Alloy Sheets"

_metals, doi:10.3390/met11122037_

Round 1

Reviewer 1 Report

This is a very interesting paper and will attract readers who are working on dissimilar metal joining. Very few researchers actually tried to measure residual stress and relate that to fatigue strength. In that respect, this paper has done very good work and deserves to be published. However, the following needs to be addressed before accepting the paper.

  1. Why different sizes were considered for tensile-shear and cross-shaped specimens. It is necessary to use the same size specimen to compare the results between tensile-shear and cross-shaped specimens.  
  2.  Additionally, it is not clear which sample was tested at 45 degrees. Was that a tensile-shear sample and cross-shaped sample or both?
  3.  Figures 6 and 7: Are they have the same condition? If yes, to have a better comparison, please merge figures 6 and 7 by putting the left half of figure 6 and the right half of figure 7 in a single figure. 
  4. Figure 8 seems unnecessary, please remove it.
  5.  It is not clear why the researchers choose to reduce the Pamp for different loading angles (Table 4)? It will be good to choose the same Pamp for the three different loading conditions. And only then, you can extrapolate the results by reducing the Pamp for different conditions.
  6. Table 4 can be moved back to the experimental section as while the reviewer was reading the experimental section it was not clear how many different loading amplitudes were tested for the fatigue life consideration.

Author Response

  1. Why different sizes were considered for tensile-shear and cross-shaped specimens. It is necessary to use the same size specimen to compare the results between tensile-shear and cross-shaped specimens.  
  2. Answer: In experimental investigations of the fatigue strength of spot joints, fatigue strength tests are usually conducted on tensile-shear, cross-tension, and coach-peel specimens. There is no international standard of the specimen geometry such as tensile-shear, cross-tension, and coach-peel specimens for evaluating the strength of SPR joints. So, we adopted and designed a cross-shaped SPR joint specimen similar to the cross-tension specimens used in spot-welded joints. A mixed-mode test jig was used for the fatigue test at load angles of 45 and 90 using a cross-shaped test specimen(Fig. 3). For load angle of 0, we adopted the tensile-shear specimens as we mentioned. The size and geometry of the tensile-shear specimen for spot-welded joint is specified by JIS(Z-3140). So, we adopted JIS standard for tensile-shear loading. The size and geometry of the tensile-shear specimen used in this study is according to JIS Z-3140. So, the size of the cross-shaped and tensile-shear specimens are different.
  3. Additionally, it is not clear which sample was tested at 45 degrees. Was that a tensile-shear sample and cross-shaped sample or both?“for the cross-shaped SPR specimens at loading angles of 45° and 90°, a special fixture, as shown in Fig. 3, was applied. This fixture is similar to the fixture for spot-welded joints proposed by Lee et al. [21]. The detailed loading fixture of the specimen is available in the literature [22]. The loading angle of 0o with cross-shaped specimen is identical to the tensile-shear loading. Thus, the tensile-shear specimen was adopted for evaluating a loading angle of 0o.”
  4. Answer: As mentioned 2.1 section, we used the the cross-shaped SPR specimens at loading angles of 45° and 90.
  • We modified the tensile-shear specimen with an applied load direction in Fig. 2(a) in order to make it easier to understand the tensile-shear loading.
  •  
  1. Figures 6 and 7: Are they have the same condition? If yes, to have a better comparison, please merge figures 6 and 7 by putting the left half of figure 6 and the right half of figure 7 in a single figure.  
  2. Answer: Please, refer to # 4 answer.
  3. Figure 8 seems unnecessary, please remove it.
  4. Answer: Figure 7 shows a detail cross-section model of the SPR joint. And, Figure 8 is a model for the specimen geometry including the SPR joint. We think that Fig. 8 is definitely necessary to explain the specimen FE modeling procedures. So, we combined Figure 7 and Figure 8.
  5. It is not clear why the researchers choose to reduce the Pamp for different loading angles (Table 4)? It will be good to choose the same Pamp for the three different loading conditions. And only then, you can extrapolate the results by reducing the Pamp for different conditions.As you know, if an fatigue experiment is conducted with a constant load using a specimen with a constant cross-section, this fatigue experiment is considered to be a stress-controlled fatigue experiment. However, for a specimen including SPR joints and spot-welded joints, it is difficult to predict their fatigue lifetimes when the thickness, shape (loading type), joint diameter of specimens are different even with the identical material. In most papers, only load amplitude-lifetime results have been reported. In order to apply their experimental results to the actual vehicle body design, fatigue lifetime must be represented by stress or other fatigue strength parameters. It is objective of this study. Therefore, the FEM structural analysis was conducted to obtain stress distribution or fatigue strength parameters under the experimental loads of these specimens.
  6. Most papers on SPR fatigue strengths have been published only with load amplitude-lifetime. However, if the thickness of the sheet, rivet diameter, and shape of the specimen (=loading type) are different, it is impossible to use these published paper data for predicting fatigue lifes even though the specimen material is identical. The SPR joint on a vehicle body panel is subject to multiaxial stress during running the vehicle. There are several fatigue parameters to predict the fatigue lifetimes of components under multiaxial stresses. That is, if the SPR joint on the vehicle body is the most vulnerable site, and if von-Mises stress on the SPR joint is known to be an appropriate parameter for fatigue lifetime prediction, the von-Mises stress amplitude-fatigue lifetime data of the specimen for these materials can be used to predict the fatigue life. When the vehicle body sheet of the identical material is joined with SPR rivets with different sheet thicknesses and diameters, we can adjust the number and position of the SPR joints on the vehicle body panel after obtaining the von-Mises stress distribution near the rivet via FEM structural analysis of the body including these joints. We include the aim and originality of this study in the introduction. as follows.
  7. Answer: The purpose of this study is not to compare fatigue strengths for three load angles, but to derive a relationship between fatigue strength and fatigue lifetime at various loading angles or complex loads. Through this relationship, we try to derive a method of predicting the fatigue strengths of the SPR joints of vehicle body during the body design.

“Most of these studies have evaluated the fatigue strengths of SPR joint specimens using load amplitude - lifetime curves instead of conventional stress - lifetime curves. However, it is difficult to predict the fatigue lifetimes of the specimens if the geometry and size of the specimens are different. Few studies have evaluated the fatigue strengths of SPR joints using structural parameters.”

“The fatigue strengths of the SPR joints under various load conditions were evaluated using various fatigue parameters to derive an appropriate structural parameter for predicting their fatigue lifetimes.”

  1. Table 4 can be moved back to the experimental section as while the reviewer was reading the experimental section it was not clear how many different loading amplitudes were tested for the fatigue life consideration.
  2. Answer: There is no regulation on the minimum numbers of fatigue testing amplitude. As you knew, if we have more testing specimens, the fatigue results will be more reliable. Table 4 is not testing procedure nor testing method. Normally, fatigue testing results will be presented in the experimental result section. So, we want to present Table 4 in the experimental result section. Please, consider our decision.

Reviewer 2 Report

  1. Clearly state the novelty aspects of the paper;
  2. What is the aim of the work?
  3. Description of production process parameters for AZ31 and Al5052 plates should be added to the manuscript.
  4. What was the distribution of residual stresses in AZ31 and Al5052 plates with thickness 1.5 mm? These results should be added.
  5. The abbreviation “SEM” does not mean “scanning electron micrograph”. It is scanning electron microscopy.
  6. How many X-ray residual stress measurements were performed for locations 1 and 2 to determine the given average values?
  7. Why the authors have not performed the testing of the microstructure in the analyzed points shown in Figure 6? These results would be interesting.
  8. Conclusions should highlight the new findings of the paper in comparision to the state of the art.

Author Response

  1. Clearly state the novelty aspects of the paper;Most papers on SPR fatigue strengths have been published only with load amplitude-lifetime. However, if the thickness of the sheet, rivet diameter, and shape of the specimen (=loading type) are different, it is impossible to use these published paper data for predicting fatigue lifes even though the specimen material is identical. The SPR joint on a vehicle body panel is subject to multiaxial stress during running the vehicle. There are several fatigue parameters to predict the fatigue lifetimes of components under multiaxial stresses. That is, if the SPR joint on the vehicle body is the most vulnerable site, and if von-Mises stress on the SPR joint is known to be an appropriate parameter for fatigue lifetime prediction, the von-Mises stress amplitude-fatigue lifetime data of the specimen for these materials can be used to predict the fatigue life. When the vehicle body sheet of the identical material is joined with SPR rivets with different sheet thicknesses and diameters, we can adjust the number and position of the SPR joints on the vehicle body panel after obtaining the von-Mises stress distribution near the rivet via FEM structural analysis of the body including these joints. We include the originality of this study in the introduction. as follows.
  2. Answer: We added the novelty aspects as you suggested.

“Most of these studies have evaluated the fatigue strengths of SPR joint specimens using load amplitude - lifetime curves instead of conventional stress - lifetime curves. However, it is difficult to predict the fatigue lifetimes of the specimens if the geometry and size of the specimens are different. Few studies have evaluated the fatigue strengths of SPR joints using structural parameters.”

“The fatigue strengths of the SPR joints were evaluated under various load conditions using various fatigue parameters to derive an appropriate structural parameter for predicting their fatigue lifetimes.”

  1. What is the aim of the work?As you know, if an fatigue experiment is conducted with a constant load using a specimen with a constant cross-section, this fatigue experiment is considered to be a stress-controlled fatigue experiment. However, for a specimen including SPR joints and spot-welded joints, it is difficult to predict their fatigue lifetimes when the thickness, shape (loading type), joint diameter of specimens are different even with the identical material. Because only load amplitude-lifetime results are reported. In order to apply their experimental results to the actual vehicle body design, fatigue lifetime must be represented by stress or other fatigue life parameters to be compared. It is objective of this study. Therefore, the FEM structural analysis was conducted to obtain stress distribution or fatigue life parameters under the experimental load of these specimens.
  2. We added the aim of the work as you suggested.
  3. Answer:

“The fatigue strengths of the SPR joints under various load conditions were evaluated using various fatigue parameters to derive an appropriate structural parameter for predicting their fatigue lifetimes.”

  1. Description of production process parameters for AZ31 and Al5052 plates should be added to the manuscript.
  2. Answer: We purchased the plates from a certain companies with mill specification sheets. The sheets show only mechanical properties and chemical compositions. Unfortunately, the production process parameters were not available. We conducted tensile tests for these plates. And, we reported in this study. What we can figure out is thickness and rolling direction. As you knew, magnesium has strong texture effect on mechanical properties. So, we added the following sentences.

“In the tensile test of the AZ31 magnesium alloy, the tensile strengths of the specimens in the rolling direction and in the transverse direction were found to be similar at approximately 277 MPa. On the other hand, the elongation at failure of the specimen in the transverse direction was close to 10%, which is nearly 0.6 times lower than the rolling direction value. The rolling direction of the top magnesium plate was installed parallel to the loading direction during specimen preparation.”

  1. What was the distribution of residual stresses in AZ31 and Al5052 plates with thickness 1.5 mm? These results should be added. 
  2. Answer: Residual stresses have components of σr and σθ. Figure 10 shows the residual stress (a) σr and (b) σθ distributions for the SPR joint specimen with thickness 1.5 mm.
  3. The abbreviation “SEM” does not mean “scanning electron micrograph”. It is scanning electron microscopy. 
  4. Answer: We corrected the SEM (scanning electron microscopy) as you pointed out.
  5. How many X-ray residual stress measurements were performed for locations 1 and 2 to determine the given average values? 
  6. Answer: This measurement was performed three times to calculate the average value.
  7. Why the authors have not performed the testing of the microstructure in the analyzed points shown in Figure 6? These results would be interesting. 
  8. Answer: This study is mainly focused on the fatigue strength of SPR joints where residual stresses exist. We cannot cover so many aspects in this paper. Unfortunately, the effect of of microstructures at the SPR joint on the fatigue strength was excluded. As you suggested, it seems interesting to study the effects of of microstructures at SPR joint on the fatigue strength. We will investigate the effect of hardness of the joint due to work-hardening on fatigue strength of the joint.
  9. Conclusions should highlight the new findings of the paper in comparision to the state of the art.As mentioned previously, most of SPR studies evaluated the fatigue strength of SPR joint samples using load amplitude-period curves instead of conventional stress-period curves. Therefore, it is difficult to predict their fatigue lifetimes when the thickness, shape (loading type), joint diameter of specimen is different even with the identical material. So, we evaluated the fatigue strengths of the SPR joints under various load conditions using various fatigue parameters. We derive an appropriate structural parameter for predicting their fatigue lifetimes. Finally, we found out that the equivalent stress intensity factor is found to be most appropriate. And, we confirmed that the fatigue lifetime of these two types of specimens could be appropriately predicted within a factor of three using the relationship of the fatigue lifetime and the equivalent stress intensity factor of the T.M-B.S.
  10. Answer: Conclusion 3 and 4 are definitely new findings of this paper in comparision to the state of the art.

Round 2

Reviewer 1 Report

The authors have addressed all the issues raised by the reviewers. It can be accepted now.

Author Response

We revised this manuscript according to your suggestion.

Reviewer 2 Report

Please correct "SEM (scanning electron micrography)" on SEM (scanning electron microscopy)

Author Response

(The authors gave the same response as above.)
